# Direct and Indirect Effects of Indoor Particulate Matter on Blood Indicators Related to Anemia

**DOI:** 10.3390/ijerph182412890

**Published:** 2021-12-07

**Authors:** Youngrin Kwag, Shinhee Ye, Jongmin Oh, Dong-Wook Lee, Wonho Yang, Yangho Kim, Eunhee Ha

**Affiliations:** 1Department of Environmental Medicine, School of Medicine, Ewha Womans University, Seoul KS013, Korea; 2loveletter2@hanmail.net (Y.K.); Jongminoh@ewha.ac.kr (J.O.); 2Graduate Program in System Health Science and Engineering, Ewha Womans University, Seoul KS013, Korea; 3Occupational Safety and Health Research Institute, Korea Occupational Safety and Health Agency, Incheon KS006, Korea; shinheeye@kosha.or.kr; 4Department of Preventive Medicine, College of Medicine, Seoul National University, Seoul KS013, Korea; taesanglee89@gmail.com; 5Department of Occupational Health, Daegu Catholic University, Gyeongsan-si KS002, Korea; whyang@cu.ac.kr; 6Department of Occupational and Environmental Medicine, Ulsan University Hospital, University of Ulsan College of Medicine, Ulsan KS016, Korea

**Keywords:** indoor particulate matter, housewives, Hb, MCV, MCH, MCHC, folate

## Abstract

Exposure to indoor particulate matter (PM) is a potential risk factor that increases systemic inflammation and affects erythropoiesis. This study investigated the association between exposure to indoor PM and blood indicators related to anemia (BIRA) in housewives. Indoor PM and blood folate status are important factors in the risk of anemia. This was a housewife cohort study; we recruited 284 housewives in Seoul and Ulsan, Republic of Korea. Indoor exposure to PM2.5 and PM10 was measured by gravimetric analysis and sensors. We investigated the BIRA, such as hemoglobin (Hb), hematocrit, mean corpuscular volume (MCV), mean corpuscular Hb (MCH), and mean corpuscular Hb concentration (MCHC). Statistical analysis was performed by multiple linear regression model and mediation analysis. The association between BIRA and PM was assessed by multiple linear regression models fitted by mediation analyses. The increase in the level of indoor PM2.5 was associated with a decrease in MCV (Beta coefficient (B): −0.069, Standard error (SE): 0.022) and MCH (B: −0.019, SE: 0.009) in gravimetric measurements. The increase in the level of indoor PM2.5 was associated with a decrease in Hb (B: −0.024, SE: 0.011), hematocrit (B: −0.059, SE: 0.033), and MCV (B: −0.081, SE: 0.037) and MCH (B: −0.037, SE: 0.012) in sensor measurements (PM2.5-Lag10). Further, we identified a serum folate-mediated PM effect. The indoor PM exposure was significantly associated with decreased Hb, MCV, and MCH in housewives. Taken together, our data show that exposure to indoor PM is a risk factor for anemia in housewives. Blood folate concentration can be a mediating factor in the effect of indoor PM on BIRA. Therefore, folate intake should be recommended to prevent anemia in housewives. Moreover, indoor PM exposure should be managed.

## 1. Introduction

Housewives with children spend relatively more time at home when compared with working women. As a result, they are exposed to a greater amount of indoor particulate matter (PM) through housework such as cooking, cleaning, laundry, and childcare. Many women and children are exposed to high levels of indoor toxic air pollutants when using fuel for cooking and heating at home [1]. PM exhibits various toxicities depending on its constituents. Indoor PM can increase the concentration of zinc, lead, mercury, and iodine and decrease vitamins B6 and B12, folic acid, and homocysteine, causing anemia. In addition, any increase or decrease in these molecules may adversely affect hematopoietic function [2].

Blood indicators related to anemia (BIRA) include hemoglobin (Hb), hematocrit (Hct), mean corpuscular volume (MCV), mean corpuscular Hb (MCH), and mean corpuscular Hb concentration (MCHC). Hb refers to its concentration in the blood. Hct refers to the percentage of the volume in contact with red blood cells (RBCs) in whole blood [3]. The WHO has defined Hb deficiency, the most representative of the BIRA, as <13 or <12 g/dL for men or women, respectively [4]. MCV is the mean Hb volume, MCH is the average Hb ratio per red blood cell, and MCHC is the Hb ratio per dl. MCV can be calculated as (Hct/100)/RBC; MCH as Hb/RBC; and MCHC is Hb/(Hct/100).

Exposure to air pollution can increase oxidative stress, a possible contributing factor to anemia [5,6]. Oxidative stress is a common denominator in the pathogenesis of many chronic diseases and affects hematopoiesis [5]. Furthermore, an increase in oxidative stress and systemic inflammation can affect the hematopoietic activity [7]. This makes cells more vulnerable and may decrease the production of red blood cells and hemoglobin, causing anemia [8]. Moreover, exposure to volatile pollutants induces metabolic diseases associated with white blood cells, such as leukemia [7]. Acute exposure to PM, especially PM2.5, induces changes in DNA methylation via inflammation and oxidative stress [6]. Folic acid is an anemia marker and antioxidant, which protects cells and tissues [6,9].

Prolonged exposure to PM can increase the risk of cardiovascular, respiratory, and endocrine diseases and systemic inflammation [10]. Additionally, it can increase the risk of anemia, leading to increased mortality, impaired functional status, and cognitive impairment [8]. Anemia is a serious condition among women and young children worldwide [1]. Exposure to air pollution greatly increases the incidence of anemia, which indicates it as important risk factor for anemia [11]. 

Several studies have measured the association between Hb and indoor air pollution, few have investigated the relationship between PM and anemia. More research is needed on this topic because of its increasing public health importance. Therefore, this study focused on the short- and long-term effects of exposure to PM in housewives, using BIRA. We established a “Housewife Cohort” to examine the association between exposure to PM and the health of housewives. We hypothesized that increased exposure to indoor pollution would adversely affect the BIRA of women and cause anemia. 

## 2. Materials and Methods

We recruited mothers from an ongoing cohort study, the Korean Children’s Environmental Health Study (Ko-CHENS). The specific cohort profile has been previously described [12]. A PM survey on the health of housewives was conducted at the University of Ulsan, South Korea, during the study period (2018–2020). The study protocol was approved by the Institutional Review Board of Ulsan University (IRB file number: 2017-12-013-009). All participants provided written informed consent [13].

### 2.1. Participants 

In this study, 284 participants were recruited from the 319 housewives of Ko-CHENS, aged between 20 and 50 years during the observation period of 2018–2020 (Figure 1).

Thirty-five participants were excluded from the study, including 23 who had missing data and 12 who did not participate in the study for ≥15 days after PM was measured. Only those housewives who reported residing indoors for a long period of time every day were selected for this study. Therefore, 284 participants were finally enrolled in the study. The average age was 34 years. All participants were married housewives living indoors with childcare responsibilities. The average number of children was 1.59, with an average 3.28 years of childcare after giving birth. We recorded the BMI, smoking frequency, and alcohol consumption for all participants once as part of the short-term study. These lifestyle habits were measured twice a year for participants who took part in the long-term study (*n* = 100) (Figure 2). Smoking and frequent alcohol consumption were reported in 10% of participants. Blood samples were collected 2–3 days before PM measurements were conducted using gravimetric analysis and sensors.

A clinical nurse collected the blood from most participants at hospital; once for the short-term study and twice for the long-term study. Participants were in a fasting state, and blood (25 mL) and urine (15 and 50 mL) were collection. Whole blood and serum and urine were collected in EDTA tubes and stored at 4 °C. White blood cells (WBCs), RBCs, Hb, Hct, MCV, MCH, MCHC, and platelets were analyzed by flow cytometry using the analysis equipment, ADVIA2120i (Siemens, Spring House, PA, USA). Folic acid and ferritin, major variables associated with anemia, were tested using the CIA analysis method with ADVIA Centaur XP (Siemens, USA) analysis equipment. Trans,trans-Muconic acid (t,t-MA) was assessed in the urine samples using high performance liquid chromatography-triple tandem mass spectrometry (HPLC-MS/MS). To maintain standardization, all samples were analyzed through SCL, a certified company [14].

### 2.2. Evaluation of PM Concentration

Direct measurements of indoor PM exposure were conducted through using the gravimetric analysis, sensors, light scattering method, and outdoor monitoring. In addition, PM concentration was repeatedly measured based on the period and life activities of the housewife. Gravimetric analysis was performed twice a year during the study period, and the sensor method was used continuously. Outdoor monitoring data were collected from the Korea Meteorological Administration. PM concentrations were compared and the exposure factors were identified. The effect of exposure to PM on the health was confirmed by using questionnaires and analysis of blood/urine samples. The addresses of all the participants were collected while determining the outdoor PM values, which were then compared with those published by the Air Korea Monitoring Network. Exposure values were corrected according to the exposure formula, as described previously [15,16].

Gravimetric analysis was performed during continuous PM exposure over one 24 period. Briefly, PM was collected on filter paper, which was first weighed at 20 °C and 50% relative humidity before exposure. First, we checked for any contamination in the separator and the normal operation of the collector. Next, the PM sample was collected on the filter paper by fixing it to the holder so that air did not leak (flow rate = 5 L/min). As per indoor air quality process test standards, we selected a place > 1 m away from the inner wall, ceiling, and floor surface (a relatively large living room in the house) and used a 1.0–1.5 m tall tripod to install the PM equipment (PM10 and PM2.5). If the house was not spacious or well-organized, PM was measured by installing the equipment at an appropriate position as deemed fit by the measurer. This is currently considered one of the most reliable methods for measuring PM concentration. After collecting the sample for 24 h, the filter paper was weighed and the concentration of PM in the indoor air was measured by determining the difference in the weight of the filter paper before and after collection. 

We used a light scattering sensor and the Internet of Things (IoT) to convert real-time measured values and concentration values into a DB for 1 year per ID (study participants) continuously to determine the extent to which housewives were exposed to PM (Kim, 2004; Shi, 2017). Additionally, we used the Community Multiscale Air Quality (CMAQ) model, a computational tool for measuring air quality data, to reproduce the seasonal PM concentration field (1-km grid resolution) for 1 year. The outdoor PM concentration was corrected using the CMAQ, and the concentration estimate of the unmeasured point was improved by considering the average outdoor concentration reproduced in the model. The outdoor sensor values were evaluated for integrated PM exposure via time-weighted average concentrations based on the CMAQ values. The latter was calculated using the average occupancy period and average exposure concentration based on region and distribution (Equation (1)),
(1)$$\mathrm{Time}-\mathrm{weighted}\mathrm{average}\mathrm{concentration}=\frac{\sum_{i=1}^{n}tici}{\sum_{i=1}^{n}ti}$$
where *t* is the time spent in the microenvironment (h) and *c* is the concentration of air pollutants in the microenvironment.

### 2.3. Statistical Analysis

The housewife cohort construction data were extracted through multilinear regression and mediation analyses. We used the correlation between indoor and outdoor PM exposure and anemia (determined through survey, and blood and urine tests) by utilizing the PM weight method and sensor measurement data. First, weight measurements were correlated through multiple linear regression analyses. Data were adjusted for BMI, ferritin, and t,t-MA, smoking, alcohol experience. BMI is an index related to lifestyle, it can be a confounding factor between X: indoor PM and Y: BIRA(blood index related to anemia). Ferritin is also a confounding factor of PM and anemia. T,t,MA (trans,trans-Muconic acid) is an indicator of VOC metabolites. Second, long-term sensor measurement values were analyzed using multiple linear regression analyses based on the 1-year exposure values and health effects. Data were adjusted for BMI, ferritin, smoking, and alcohol experience. Third, a mediation analysis was performed according to short-term and long-term PM exposure, blood folate concentration, and BIRA without adjusted confounding variables. The effect of folate was confirmed through a mediated analysis on the correlation between exposure to PM and BIRA. Data were considered significant if *p* < 0.05. All analyses were performed using SAS (version 9.3; SAS Institute Inc., Cary, NC, USA) and R (version 3.6.0; R Foundation for Statistical Computing, Vienna, Austria).

## 3. Results 

The characteristics of the study participants are listed in Table 1. Of the 284 housewives who participated in the study, 31 (10.92%) had anemia symptoms (Hb ≤ 12 mg/dL). A comparison of the characteristics of the non-anemic and anemic groups showed that both the age and BMI–underweight ratio was higher in the anemic group than in the non-anemic group. But it was not significantly. By contrast, the Hb, hematocrit, MCV, MCH, MCHC, and ferritin concentrations were significantly lower in the anemic group than in the non-anemic group. In addition, there were a higher percentage of housewives in the anemic group who reported having alcohol experience, drinking more than once a week, non-exercisers, educated up to high school or lower in Table 1. It was also noted that a higher proportion of housewives in the anemic group used electricity for cooking. But they were not significant, too. 

The association between gravimetric PM2.5 and PM10 and BIRA is presented in Table 2. Significant reductions were observed in MCV, MCH after exposure to PM2.5 and PM10. Data were adjusted for BMI, ferritin, t,t-MA, smoking, alcohol experience following the gravimetric method. The results indicated that the BIRA have a negative correlation with PM_2.5_ and PM_10_ concentrations.

The results of long-term exposure on the BIRA are listed in Table 3. The monthly exposure period was divided into lag01 (monthly exposure) and lag012 (12-month exposure) periods based on sensor measurement values for 1 year. Hb, hematocrit, MCV, and MCH were significantly decreased after 9–11 months of exposure, while RBCs showed a significant decrease in lag011 PM2.5 alone. These results indicated a negative correlation between PM_2.5_ sensor exposure values and the BIRA. 

A significant correlation was observed by mediation analysis between gravimetric PM10 and blood folate, MCH, and MCHC in Table 4. Exposure to PM decreased MCV, MCH, and MCHC. Furthermore, it decreased the concentration of folic acid in the blood (Figure 3).

Table 5 shows the correlation between PM2.5, blood folic acid, and BIRA through mediation analysis. A significant negative correlation was observed between sensor PM2.5(Lag10) and blood folate, MCV and MCH; PM reduced MCV, MCH, and blood folate concentrations. Parametric analysis was performed to analyze the relationship between the three variables of PM (X) and the BIRA (Y), using folic acid as a parameter. Factors X and Y correlated with blood folate levels. It is very meaningful that the mediation analysis on long-term exposure to fine dust showed significant results in the same index as the mediation analysis on short-term exposure.

## 4. Discussion 

### 4.1. Mechanism of PM Action

Acute exposure to PM generally induces alterations to DNA methylation based on inflammation and oxidative stress. Once inhaled, PM causes inflammation by creating oxidative stress in the lungs and triggering the production of main species of ROS (such as peroxides, hydroxyl radicals, and nitric oxide), adversely affecting red blood cells and causing anemia [1,6,10]. In particular, PM2.5 induces changes in the methylation of genes involved in mitochondrial oxidative energy metabolism. However, vitamin B supplementation can prevent such damage [17,18]. Furthermore, the effect on mortality persists even after several decades of exposure to PM, similar to smoking [10,17].

Most studies have actively investigated PM on iron deficiency and another anemia [3,8,19]. Serum folate concentration < 6.6 nmol/L (<3.0 ng/mL) is considered as folate deficiency, cause of anemia [9]. Blood folic acid is a blood indicator related to anemia and is also a component related to oxidative stress [5,6]. Folate deficiency disrupts the hematopoietic process, which can be assessed as a change in the erythrocyte index [5,8]. Folate deficiency can lead to an abnormal increase in MCV. However, a recent study showed that folate-deficient patients may have normal-cell (23.7%), small-cell (11.1%), or macrocytic (2.0%) anemia. As a result, folate deficiency is associated with abnormal shape and quality of blood cells [20,21].

### 4.2. PM and Health Effects

Exposure to PM can increase the concentration of systemic inflammatory markers, which *are associated with* cardiovascular disease [10,22]. An increase in inflammatory markers is an indicator of respiratory diseases. They can decrease base-pairing error repair proteins in airway epithelial cells, which causes an increase in the incidence of cardiovascular-disease risk factors; endocrine diseases, such as diabetes and hypertension; female reproductive system diseases, such as infertility; and neuropsychiatric symptoms, such as anxiety [8,10,23]. Previous studies on Hb and the prevalence of anemia have shown an association between PM2.5, nitrogen dioxide (NO2), and access to blood (plasma and forming cell elements) [8,24]. The authors have argued that it makes them exceptionally vulnerable to the harmful effects of possible contaminants, such as lead and arsine, which damage the membranes of erythrocytes and cause anemia [7,25].

Previous studies have indicated an association between low-folate status and anemia [20]. Although reductions in long-term exposure to air pollution are associated with increased survival, it has been suggested that the effects on mortality persist even decades after exposure, similar to those observed with smoking [10,17,26]. Additionally, the effects of exposure to PM on serum, RBC, and folate levels have been confirmed by weak evidence from previous studies [1]. One previous study has reported that the use of biofuels (wood, feces, and crop residues) for cooking and heating affects the association between anemia and stunting in children in families [5,25,27]. Another study has showed that pulmonary oxidative stress in response to inhalation exposure (active or second-hand tobacco smoke exposure) or air pollution is a major factor for inflammation and ROS-induced oxidative tissue damage. Moreover, a significant positive correlation has been observed between blood folate and Hb concentration [5,17,28]. This suggests that folate deficiency limits Hb biosynthesis. A previous study that used a model to detect the potential predictors of Hb concentration has demonstrated that folic acid status influences Hb concentration [1,6]. Interestingly, studies have shown significant public health benefits in areas with high PM2.5. Therefore, interventional prevention (vitamin intake) at the individual level is needed to control the side effects of PM2.5 and supplement the regulators of this variable [6]. 

Oxidative stress is a common denominator in the pathogenesis of many chronic diseases; therefore, antioxidants are frequently used to protect cells and reverse oxidative damage. Antioxidants mediate hematopoietic homeostasis and oxidative stress, which supports our findings [5,29]. Folic acid supplementation can reduce the oxidative damage caused by PM by regulating the expression of oxidative stress and pro-inflammatory methylation [6,30]. Hence, individual-level prevention can be used to control and complement the potential pathways of anemia and hematopoietic mechanisms caused by PM2.5-induced oxidative stress [6,9].

This suggests that the folate pathway is an essential part of the versatile cellular mechanism that responds to oxidative stress, and plays a crucial role in fundamental cellular processes [18]. Furthermore, exposure to air pollution is significantly associated with increased anemia and decreased Hb levels in vulnerable groups [19]. This indicates that exposure to chronic air pollution is an important risk factor for anemia in vulnerable populations [19,24]. In addition, studies have proved the relationship between pollutants, increased oxidative stress, decreased blood folic acid, and anemia [9,22,31].

### 4.3. Contribution of Our Study

Few studies have investigated the effect of indoor air pollution on anemia, although the investigation of chronic effects on blood health is increasing [1,32,33]. Current research is limited to assessing the relationship between fuel consumption and anemia in developing countries [4,32], overall outdoor air and anemia [8,11], and the association between anemia in children [34,35].

Our study shows strong short- and long-term effects to exposure to PM, similar to those reported by studies on the effects of smoking. It is difficult to investigate the relationship between indoor air pollution and health effects because of the difficulties in isolating the temporal and spatial fluctuations of exposure, different chemical compositions of indoor air pollution, study groups of different ages, and other environmental and individual risk factors. Our study obtained the same results by simultaneously executing a short-term indoor study and a long-term study under several constraints. Hence, our study shows reliable results. In addition, the mediation analysis was performed to provide evidence for the mediating effect of blood folic acid [36,37,38].

This study diversified the design for both short- and long-term studies on exposure to indoor and outdoor PM in housewives according to the measurement period. We evaluated their health effects through mediation analysis. The results of indoor and outdoor PM exposure data were compared and presented using various measurement methods. We evaluated the health effects of indoor PM on housewives by correcting for actual exposure; we constructed individual exposure values that considered both indoor and outdoor exposure, rather than by using simple measurement values. 

### 4.4. Limitations of This Study

The sample size was very small and statistical power was limited. This was due to difficulties in recruiting participants, and accurately measuring exposure and outcome variables. The exposure data estimated by the modeling method had a relatively high percentage of missing data because of limitations in the IoT sensors. Development of precision PM sensor technology and the continuity of communication connections remain important challenges. In addition, the data were related to Korean housewives alone; therefore, the results of this study cannot be generalized [20].

## 5. Conclusions

This study focused on the short- and long-term effects of exposure to PM and BIRA in housewives. We believe that this study will help to advance the development of individual exposure-assessment techniques for PM. Identifying the mechanism underlying the risk of short- and long-term exposure to PM will enable further research and help to build a risk assessment system. Additionally, this study will expand the development of the PM exposure measurements and information platforms. Moreover, it has provided scientific information on the health effects of indoor PM on housewives, which will relieve public anxiety regarding this problem.

This study can serve as a baseline to investigate the effect of interventions on public health because it offers a detailed scientific analysis of PM exposure for women with indoor lifestyles. The health of housewives is closely related to the health of infants and children; therefore, this study is also useful for the overall health management of households. The expansion of our study can support public health management policies and prepare countermeasures to reduce the exposure to PM in different daily life environments. Finally, it can help to construct actual PM exposure assessment data based on domestic living patterns and indoor pollution.

## Figures and Tables

**Figure 1 ijerph-18-12890-f001:**
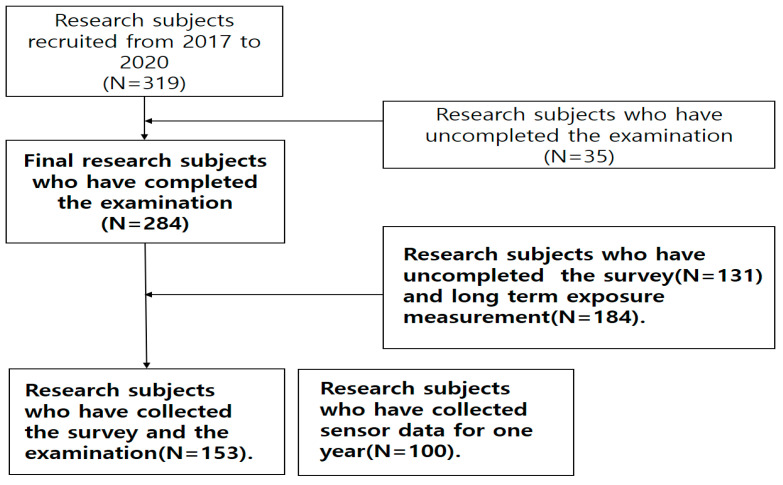
Flow chart of the study population.

**Figure 2 ijerph-18-12890-f002:**
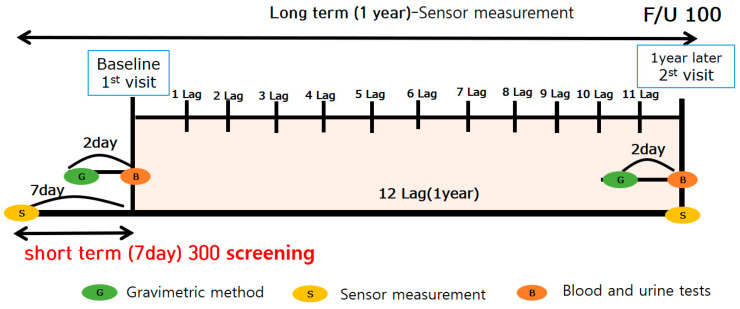
Schematic diagram of the study period.

**Figure 3 ijerph-18-12890-f003:**
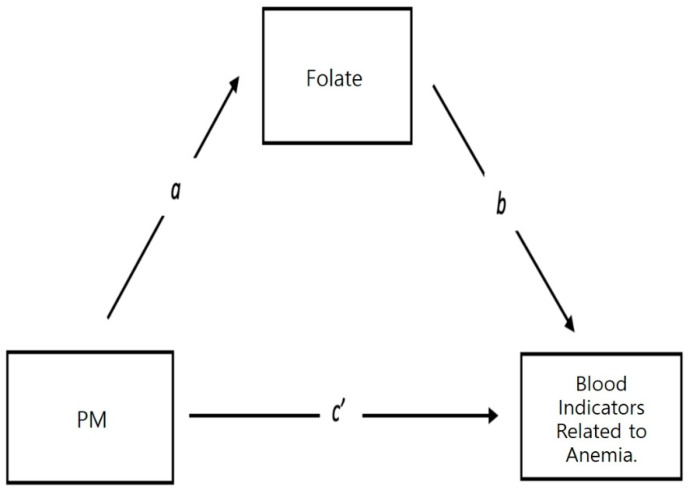
Direct and indirect effects of particulate matter and blood indicators related to anemia.

**Table 1 ijerph-18-12890-t001:** Characteristics of study subjects.

**Characteristics**	**Total Observations-284**	**Anemic-31**	**Non-Anemic-253**	***p*-Value**
	**% or Mean ± SD**			
Age (mean ± SD)	34.34 ± 3.65	35.03 ± 3.10	34.25 ± 4.59	0.245
BMI kg/m^2^ (mean ± SD)	23.43 ± 4.47	22.36 ± 3.45	23.55 ± 3.67	0.061
BMI < 18.5 (*n*, %)	20 (7.04)	3 (9.68)	17 (6.72)	0.130
18.5 ≤ BMI < 25 (*n*, %)	184 (64.79)	24 (77.42)	160 (63.24)	
BMI ≥ 25 (*n*, %)	80 (28.17)	4 (12.90)	76 (30.04)	
**PM Value**				
**Gravimetric PM2.5**	30.68 ± 20.50	28.09 ± 26.93	30.99 ± 21.50	0.565
**Gravimetric PM10**	48.12 ± 27.10	46.61 ± 37.98	48.30 ± 26.5	0.811
**Biomarker (mean ± SD)**				
Hemoglobin	13.08 ± 1.06	11.13 ± 0.83	13.3 ± 0.82	<0.001
Hematocrit	40.43 ± 3.04	35.4 ± 2.02	41.01 ± 2.56	<0.001
MCV	92.74 ± 4.92	88.68 ± 8.43	93.21± 4.11	<0.001
MCH	30.0 ± 1.86	27.90 ± 3.33	30.25 ± 1.42	<0.001
MCHC	32.34 ± 0.89	31.39 ± 1.05	32.45 ± 0.80	<0.001
Folate	11.16 ± 7.07	10.25 ± 4.21	11.27 ± 7.35	0.301
Ferritin	43.01 ± 33.59	9.41 ± 7.18	47.15 ± 33.61	<0.001
***p*-Value; *t*-test**				
**Characteristics**	**Total Observations** **(N = 153)**	**Anemic** **(N = 18)**	**Normal** **(N = 135)**	***p*-Value**
	**% or Mean ± SD**			
Age (mean ± SD)	34.42 ± 3.60	35.11 ± 3.48	34.33 ± 3.38	0.377
BMI kg/m^2^ (mean ± SD)	23.02 ± 3.39	22.54 ± 3.85	23.08 ± 3.58	0.576
BMI < 18.5 (*n*, %)	11 (7.19)	3 (16.67)	8 (5.93)	0.252
18.5 ≤ BMI < 25 (*n*, %)	103 (67.32)	11 (61.11)	92 (68.15)	
BMI ≥ 25 (*n*, %)	39 (25.49)	4 (22.22)	35 (25.93)	
**PM Value**				
**Gravimetric PM2.5**	28.32 ± 16.99	23.02 ± 12.26	29.02 ± 17.44	0.076
**Gravimetric PM10**	44.76 ± 21.89	38.33 ± 15.27	45.62 ± 22.53	0.085
**sensor PM2.5**	11.63 ± 7.67	11.91 ± 6.98	11.59 ± 7.78	0.857
**sensor PM10**	21.69 ± 13.82	21.90 ± 12.16	21.66 ± 14.06	0.939
**Biomarker (mean ± SD)**				
Hemoglobin	12.95 ± 0.97	11.25 ± 0.49	13.18 ± 0.77	<0.001
Hematocrit	40.17 ± 2.87	35.85 ± 1.37	40.75 ± 2.50	<0.001
MCV	93.05 ± 4.58	89.26 ± 6.81	93.56 ± 3.96	0.017
MCH	30.02 ± 1.72	28.01 ± 2.50	30.29 ± 1.39	0.001
MCHC	32.25 ± 0.79	31.36 ± 0.74	32.37 ± 0.71	<0.001
Folate	11.21 ± 7.07	11.00 ± 4.21	11.23 ± 7.35	0.849
ferritin	40.18 ± 33.59	9.412 ± 7.18	43.97 ± 33.62	<0.001
**Alcohol** **experience**				
No (*n*, %)	44 (28.76)	3 (16.67)	41 (30.37)	0.177
Yes (*n*, %)	109 (71.24)	15 (83.33)	94 (69.63)	
**(Drinking more than** **once a week)**				
No (*n*, %)	97 (88.99)	13 (86.67)	84 (89.36)	0.470
Yes (*n*, %)	12 (11)	2 (13.33)	10 (10.63)	
**Physical activity**				
No (*n*, %)	124 (81.05)	15 (83.33)	109 (80.74)	0.792
Yes (*n*, %)	29 (18.95)	3 (16.67)	26 (19.26)	
**Education level**				
≤High school (*n*, %)	32 (20.92)	6 (33.33)	26 (19.26)	0.168
≥college (*n*, %)	121 (79.08)	12 (66.67)	109 (80.74)	
**Smokingexperience (*n*, %)**				
Yes	23 (15.03)	2 (11.11)	21 (15.56)	0.620
**Cook fuel**				
gas (*n*, %)	102 (66.67)	10 (55.56)	92 (68.15)	0.287
**electricity (*n*, %)**	51 (33.33)	8 (44.44)	43 (31.85)	
***p*-Value; *t*-test,** **Survey *p*-Value; chi-squared test**				

Hb, hemoglobin; MCV, mean corpuscular volume; MCH, mean cell Hb; MCHC, mean corpuscular Hb concentration.

**Table 2 ijerph-18-12890-t002:** Regression model of baseline particulate matter (gravimetric method) for blood indicators related to anemia.

Blood Indicators Related to Anemia	Gravimetric PM2.5			Gravimetric PM10		
	Estimate	Std. Error	Pr (>|*t*|)	Estimate	Std. Error	Pr (>|*t*|)
RBC	0.003	0.002	0.158	0.002	0.002	0.224
Hb ^1^	0.001	0.005	0.905	0.000	0.005	0.946
Hematocrit	−0.003	0.015	0.833	−0.004	0.014	0.789
MCV ^2^	**−0.069**	**0.022**	**0.003 ****	**−0.064**	**0.022**	**0.005 ****
MCH ^3^	**−0.019**	**0.009**	**0.039 ****	**−0.019**	**0.009**	**0.036 ****
MCHC ^4^	0.003	0.004	0.401	0.002	0.004	0.635

Multiple linear regression model, adjusted for BMI, ferritin, t,t-MA, Smoking, Alcohol experience. ** *p* < 0.05; ^1^. Hb, hemoglobin; ^2^. MCV, mean corpuscular volume; ^3^. MCH, mean cell Hb; ^4^. MCHC, mean corpuscular Hb concentration.

**Table 3 ijerph-18-12890-t003:** Regression model of particulate matter (sensor-long-term exposure) for blood indicators related to anemia.

Blood Indicators Related to Anemia	RBC ^1^	Hb ^2^	Hematocrit	MCV ^3^	MCH ^4^	MCHC ^5^
Moving average	Estimate (Std. Error)	Estimate (Std. Error)	Estimate (Std. Error)	Estimate (Std. Error)	Estimate (Std. Error)	Estimate (Std. Error)
Sensor PM2.5						
lag01	−0.002	0.011	−0.005	0.037	0.038	0.027
	(0.009)	(0.024)	(0.081)	(0.074)	(0.025)	(0.024)
lag02	0.004	0.018	0.029	−0.013	0.010	0.015
	(0.009)	(0.025)	(0.082)	(0.093)	(0.031)	(0.026)
lag03	0.013	0.040	0.070	−0.115	−0.009	0.033
	(0.010)	(0.029)	(0.094)	(0.106)	(0.038)	(0.031)
lag04	−0.004	−0.007	−0.043	−0.019	0.007	0.014
	(0.007)	(0.020)	(0.064)	(0.072)	(0.025)	(0.020)
lag05	−0.005	−0.011	−0.031	0.038	0.011	−0.003
	(0.006)	(0.016)	(0.051)	(0.058)	(0.021)	(0.016)
lag06	−0.003	−0.012	−0.023	0.025	−0.002	−0.013
	(0.005)	(0.015)	(0.044)	(0.052)	(0.019)	(0.015)
lag07	0.000	0.005	0.014	0.040	0.011	−0.001
	(0.004)	(0.011)	(0.034)	(0.041)	(0.014)	(0.012)
lag08	0.000	0.000	−0.017	−0.043	−0.002	0.013
	(0.004)	(0.014)	(0.043)	(0.047)	(0.017)	(0.015)
lag09	−0.006	**−0.034 ****	**−0.105 ****	**−0.112 ****	**−0.039 ****	−0.002
	(0.004)	**(0.012)**	**(0.037)**	**(0.047)**	**(0.016)**	(0.014)
lag010	−0.003	**−0.024 ****	**−0.059 ***	**−0.081 ****	**−0.037 ****	−0.011
	(0.004)	**(0.011)**	**(0.033)**	**(0.037)**	**(0.012)**	(0.011)
lag011	**−0.009 ****	**−0.033 ****	**−0.086 ****	−0.016	−0.015	−0.011
	**(0.004)**	**(0.012)**	**(0.037)**	(0.032)	(0.012)	(0.013)
lag012	0.006	0.011	0.032	−0.041	−0.015	0.000
	(0.006)	(0.018)	(0.062)	(0.052)	(0.020)	(0.023)

Multiple linear regression model, adjusted for BMI, ferritin, Smoking, Alcohol experience. lag01(lag0–1): 1 month exposure. lag012(lag0–12): 12 months of exposure. * *p* < 0.1; ** *p* < 0.05; ^1^. RBC, red blood cell; ^2^. Hb, hemoglobin; ^3^. MCV, mean corpuscular volume; ^4^. MCH, mean cell Hb; ^5^. MCHC, mean corpuscular Hb concentration. in bold-Estimate (Std. Error).

**Table 4 ijerph-18-12890-t004:** Mediation analysis of baseline study particulate matter (gravimetric PM10) exposure for blood indicators related to anemia.

Blood Indicators Related to Anemia	a	b	a × b	c’	c
			(Indirect Effect)	(Direct Effect)	(Total Effect)
MCV	−0.017	0.081 **	−0.001	−0.007	−0.008
MCH	−0.017	0.037 **	−0.001	−0.008 *	−0.008 **
MCHC	−0.017	0.012 *	0.000	−0.006 ***	−0.006 ***

* *p* < 0.1; ** *p* < 0.05; *** *p* < 0.01. Mediation analysis model. a = PM- > folate; b = folate > blood indicators related to anemia; c = total effect of X on Yc = c’ + ab; c’ = the direct effect of X on Y after controlling for M; c’ = c − ab; ab = indirect effect of X on Y.

**Table 5 ijerph-18-12890-t005:** Mediation analysis of long-term PM exposure (sensor PM2.5−lag10) for blood indicators related to anemia.

Blood Indicators Related to Anemia	a	b	a × b	c’	c
			(Indirect Effect)	(Direct Effect)	(Total Effect)
MCV	−0.141 *	0.164 **	−0.023	−0.067	−0.090 *
MCH	−0.141 *	0.061 **	−0.009	−0.031 *	−0.039 **
MCHC	−0.141 *	0.006	−0.001	−0.01	−0.011

Note: * *p* < 0.1; ** *p* < 0.05. Mediation analysis model. a = PM- > folate. b = folate > blood indicators related to anemia; c = total effect of X on Yc = c’ + ab; c’ = the direct effect of X on Y after controlling for M; c’ = c − ab; ab = indirect effect of X on Y.

## Data Availability

Not applicable.

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
