# Peer review of "Direct and Indirect Effects of Indoor Particulate Matter on Blood Indicators Related to Anemia"

_ijerph, 2021, doi:10.3390/ijerph182412890_

Round 1

Reviewer 1 Report

Dear Editor and Authors,

The authors have significantly improved their manuscript, but two issues still remain unclear. Namely, inaccurate citations of the results of other authors studies and errors in the numbering of references, despite the authors' assurances that the citations will be corrected. The second issue concerns the description of the course of the study and the interpretation of the results of statistical tests.

Clarification of the above comments and supplementation of the manuscript will significantly increase the quality of its preparation and its comprehensibility for potential readers.

These comments are detailed below.

Line 30:

This sentence is incomplete, there is no information to which group of measurements it relates. I suppose I should have information: in sensor measurement.

Line 62:

In contexts of Leukemia, “blood cells” should be replaced by : white blood cells” or “leucocytes”.

Lines 88-110:

I strongly recommended to create a flow chart of the study with information of the number and causes of the participants excluding, and the time and sequence of performing individual determinations and measurements.

In its current form, with Table 1-2 appended, the study flow is difficult to understand.

Lines 172-180:

In chapter 2.3, the authors indicate that for the statistical tests used, p <0.05 was considered significant. Please explain why in Chapter 3 the authors still indicate a higher BMI-underweight ratio if they did not perform the chi 2 test for this comparison and a higher percentage of housewives in the anemic group who reported being Alcohol experience, Drinker more than once a week, non- exercisers, educated up to high school or lower, if the p-value is above 0.05.

Based on the chi 2 test, there are no grounds for adopting such an interpretation of the observed frequencies.

Line 177 and Table 1-2:

misspelled word: alcohol

Lines 267-268:

“ As a result, folate deficiency causes an abnormal increase in 267 blood cells [8, 14].”

- what does this sentence mean? increase in volume or count?

- conclusion from Reference 8 consider only mucosal disease patients, not general population or women. Please be meticulous in quoting the results of other authors.

- I do not find any information about the relation of folate and abnormal blood cells in reference 14.

Lines 284-285:

The Authors of the reference 27 did not assessed the PM exposure!!!

Lines 291-292

This publication [26] is about: Reduction of DNA mismatch repair protein expression in airway epithelial cells of premenopausal women chronically exposed to biomass smoke, only. There are no words “folate” and “anemia” in this publication.

Lines 293-295:

I encourage researchers to take a critical approach to the publications of other authors. It is true that Ndiaye NF et al. [27] showed a significant association of folic acid with hemoglobin concentration, but this correlation was less than very weak (r = .07), which is not important from the clinical point of view.

Line 315:

I am really confused.

Reference [12] do not concern the air pollution!

Reviewer 2 Report

The authors answered my queries. I approve the manuscript.

Author Response

Thank you.

This manuscript is a resubmission of an earlier submission. The following is a list of the peer review reports and author responses from that submission.

Round 1

Reviewer 1 Report

Review “Direct and Indirect Effects of Indoor Particulate Matter on Anemia Indicator Level”.

The authors  investigated the association between exposure to indoor PM and indicators of anemia (see below) in housewives. This is an interesting topic in the context of protecting the health of women and mothers. Both short-term and long-term exposure measurements were planned using a variety of measurement tools to assessed exposure to indoor pollutants. However, the method of describing the research methodology, the terminology used, and most of all, the failure to take into account important factors influencing the obtained results lead to drawing conclusions that raise reservations.

Specific comments are provided below:

Tilte: The authors misinterpret the concepts of MCV, MCH and MCHC red cell indicators. Red cell indicators describe the morphology of a statistical red blood cell, but are not used to diagnose anemia (see: https://medlineplus.gov/lab-tests/red-blood-cell-rbc-indices/). They are used to differentiate anemia and may indicate the cause of it. The diagnosis of anemia is based on a decrease in hemoglobin concentration and/or red blood cell counts. Therefore, the use of anemia indicator term for MCV, MCH and MCHC in the title and manuscript is unjustified.

2.1  Participants

The authors describe in detail the methodology of measuring exposure to pollution, however, the issue of the characteristics of the study group and the methodology of measuring laboratory parameters are limited only to the information in line 114 (section 2.2 Evaluation of PM2.5 concentration) about blood sampling 2-3 before the gravimetric measurement.

Section 2.1 should provide information on factors potentially influencing the development of anemia, in particular:

- age structure of participants

- frequency of smoking

- being pregnant, number of children and time since last birth

It is necessary to add information, e.g. as a separate chapter with a description of the method of obtaining blood samples and the laboratory methods used, in particular:

  - at what time of the observation and by whom the blood samples were taken, once at the beginning of the study or more frequently. This cannot be deduced from the manuscript. If the study lasted 1 year, was it checked whether the participants developed or recovered from an anemia during the follow-up? This is very important because drawing conclusions about anemia association based on one measurement, e.g. at baseline, is unreasonable.

- whether the samples were taken under fasting condition or not. Standardization of blood collection  is very important. The meal has an influence on the laboratory test results.

  - on which analyzer the completely blood count (CBC) and laboratory determinations were made

  - for what purpose the ferritin was determined and by what method

- when, how often and for what purpose the BMI was measured

2.2 Evaluation of PM2.5 concentration

The chapter title only indicates PM2.5 when the chapter text also applies to PM 10. The chapter content needs to be revised or title changed.

line 114-116 The sentence should be removed.

2.3 Statistical analysis

line 149 What parameters were measured in urine???

line 152 What the abbreviations mean: t and t-MA? Abbreviations should be expanded on first use.

  1. Results

line 169 Please, explain how the number of women with anemia has been determined.

Whether it was a baseline diagnosis and whether the women developed anemia at the time of the study. There is no agreement in the number of women with anemia quoted in the text (18 (11.76%)) and the number shown in Table 1 (31 (%?)).

line 174-177 This section with participants characteristics raises major concerns with respect to the subsequent statistical analyzes. Why did the authors not pay attention to smoking, alcoholism and the consumption of alcohol and fuel used for cooking as factors increasing the risk of lower values of folic acid and the promotion of oxidative stress?

The authors themselves mention in the introduction (line 44) that fuel, and more specifically biofuel, is the main source of indoor pollution. At the same time, the authors of the study indicate that women with anemia more often use electricity to cook. It is necessary to refer to this revelation in the discussion.

Excessive alcohol consumption has a documented effect on the decreased concentration of folic acid. The authors note that among anemic women there were definitely more alcoholics. Is this  number significantly higher? A necessary condition for correct inference is taking into account alcohol consumption as a disturbing factor in the performed regression analyzes. Authors should prove that alcohol consumption is a negligible factor or a factor independent of PM exposure related to analyzed blood count parameters.

Line 178 Table 1.

The term "normal" is inadequate to the situation and should be replaced with eg. non-anemia.

The average values should be provided together with the result dispersion index, e.g. standard deviation.

The values described as Biomeasures should be assigned the units in which they were measured. The term "biomeasures" is inadequate to the analyzed blood count parameters and unnecessary in the description of the table.

Line 200-201 This sentence is incomprehensible. What does the term "health effects" mean?

4.1 Mechanism of PM action

line 247 I do not agree with the statement that folic acid is an indicator of anemia, or the intensity of oxidative stress. As the authors themselves indicate in the introduction, the indicator of anemia in women is the concentration of hemoglobin <12 g / dL. This passage needs to be rewritten.

4.2 Previous studies on fine dust and health effects

line 255 Whether the authors are sure that this title is consistent with the nomenclature?

line 281 What kind of Biomarkers or rather one biomarker if you quoted a particular concentration?

line 288-290 This sentence is ambiguous. What benefits, what action? What does it mean “ PM2.5 is frequent”? 

4.4 Limitations of the study

line 336 Please provide the exact percentage of missing data. How were they included in the analysis (e.g. mathematical complementation of missing items) or removed from the analysis (in pairs or cases).

Line 341-342 What do the authors mean when they say "blood health indicators", do they mean red cell indicators? The conclusions must be strictly related to the results obtained.

Generally, it is necessary to re-analyze the results with smoking and alcohol consumption as independent variables. Without taking these factors into account, the conclusions drawn are unreliable.

Reviewer 2 Report

This study is aimed to associate indoor air pollution with anemia in women. The research contributes to increasing our knowledge related to the importance to reduce PM2.5 in our life due to the negative impact on health. Although the study is important sometimes was difficult for me to understand completely the idea of the authors. I believe the English style should be improved. 

It is surprising for me that whereas Hb is clearly related to hematocrit values, the association of both measurements with PM2.5 was in opposed directions. The authors should assess multicollinearity. The authors should assess Hb or hematocrit but not both in the same model.

The reference is incomplete. More citations are required. I am enclosing some references.

The reference from Honda includes outdoor data but the authors say in their paper that Honda studied indoor contamination. This should be corrected.

The discussion section says that folate deficiency would be associated with low, normal, or higher RBC size, and data from the study showed that MCV is lower in the group with PM2.5 exposure. The authors should classify NCV as low, normal, and higher and then determine how is distributed these groups according to the quartile of exposure to PM2.5.

The authors should discuss the PM2.5 composition. Iron is also part of PM2.5 and this may induce inflammation and oxidative stress. This also may cause anemia with iron overload in tissues. We have included a couple of references.

It is interesting the measurement of serum folate and the demonstration that lower values are associated with PM2.5 exposure. How much megaloblastic anemia was observed in the group with high PM2.5 exposure?

Round 2

Reviewer 1 Report

In my opinion, the improvement of the understanding and correctness of the nomenclature used and the interpretation of the results made by the authors is not sufficient:
- in the first version of the manuscript, the authors testified that there were more alcoholics in the anemic group (line 182-183). Despite the fact that alcohol consumption is taken into account, the supplementary table 1 only shows whether a given person consumes alcohol or not. Drinking alcohol is one of the most important factors in influencing folic acid levels. The authors indicate that "Smoking and frequent alcohol consumption was reported in 10% of participants". Such a low percentage of people frequently consuming alcohol intake does not justify the omission of this parameter as a disturbing factor.

- for what purpose the division of participants according to the BMI classification is presented in table 1, did it have any influence on the further part of the analysis? What links may exist between BMI and red cell indices that this parameter has been included as a confounding factor?
- no introduction or justification has been made for the performance of the determinations t, t-MA. For what purpose was a regression adjustment made to this parameter?

The authors provide specific frequencies of such qualitative parameters as the presence of BMI classes (Table 1), alcohol consumption, physical activity, education level, etc. Correct interpretation of these values ​​requires the Ch2 test to be performed. e.g. for alcohol consumption the result of the chi2 test (anemia: 3/15 and non-anemia 15/94 people) is p = 0.228. This indicates that there is no reason to state that alcohol consumption is more frequent in the group of people with anemia.

- there is no reference to supplementary materials in the text

- There are unacceptable factual errors in the manuscript, e.g. the sentence (line 279-281): Furthermore, rural women have a higher concentration of folate (7.27 nmol / L [6.89–7.68 nmol / L]) than urban women (10.45 nmol / L [9.88–280 11.05 nmol / L]). The results of other authors were incorrectly cited, which additionally concern a completely different study population[37].

There are citations in the manuscript that are not reflected in the references cited.
- item 14: in this paper there is no information that oxidacin and inflammatory stress damage the erythrocyte membrane and cause anemia. Moreover, damage to the erythrocyte membrane leads to a specific type of haemolytic anemia.

- in item 32 I did not find information that Folic acid is an anemia marker and antioxidant, which protects cells and tissues

Due to the above-mentioned comments and the lack of consistency between the presented results and the conclusions drawn in terms of changing the MCV and reducing the concentration of folic acid, in my opinion, the manuscript should not be published in this form.

Reviewer 2 Report

The authors have improved substantially the manuscript. I have some minor comments:

Line 243: "nitric oxide" is repeated. Line 251: Is folate deficiency a major cause of anemia?, Reference 26 refers that folate deficiency is the minor cause of anemia after ID and Vit B12 deficiency. Please correct.

Table 1. What does mean BMI 24?

Table 1 should be included PM2.5, PM10, Ferritin, and folate data.

In table 2  is not clear what is the dependent variable and what the independent variables. As presented seems that PM2.5 and PM10 are the dependent variables. Please clarify. 

The authors have not assessed for multicollinearity as requested.
